# HERA: Efficient Test-Time Adaptation for Cross-Domain Few-Shot Segmentation with Vision Foundation Models

## Abstract

Vision foundation models (VFMs) have achieved strong performance across various vision tasks. However, it still remains challenging to apply VFMs for cross-domain few-shot segmentation (CD-FSS), which segments objects of novel classes under domain shifts using only a few labeled exemplars. The challenge is mainly driven by two factors: (1) limited labeled exemplars per novel class relative to the scale of VFM pre-training, causing overfitting prone under retraining, and (2) target-domain shifts underrepresented during pre-training, inducing cross domain inconsistency and layerwise sensitivity. To address these issues, we propose Hierarchical Exemplar Representation Adaptation (HERA), a three-stage select-regularize-calibrate VFM-based segmentation framework that learns effectively from limited labels and adapts to novel domains without source-data retraining. We first design Hierarchical Layer Selection (HLS) to adaptively identify the most informative VFM layer using a data-dependent Exemplar Transfer Risk (ETR) computed for each candidate layer. Then Prior-Guided Regularization (PGR) regularizes interactions on the selected representation, yielding well-structured local signals for the subsequent stage. Furthermore, Pixelwise Adaptive Calibration (PAC) combines the selected representation with the refined interaction maps to calibrate pixelwise predictions, producing consistent masks. Together, these stages form a hierarchical select–regularize–calibrate pipeline that guides frozen VFM features in new domains while fine-tuning less than 2.7% of parameters at test time. Extensive experiments show that HERA surpasses the state-of-the-art by more than 6.5 mIoU across multiple CD-FSS benchmarks.

## 1 Introduction

Few-shot semantic segmentation (FSS) aims to generate pixel-level predictions for novel classes using only a few labeled support exemplars. Existing methods (Wang et al., 2019; Min et al., 2021; Zhang et al., 2021; Fan et al., 2022) learn class-agnostic correspondences from constructed pairs of support and query images, which transfers knowledge from base to novel classes and yields strong in-domain generalization. However, when the deployment distribution differs from the training distribution, the learned correspondences and class prototypes degrade, leading to large performance drops. This motivates cross-domain few-shot segmentation (CD-FSS), which seeks to generalize to novel classes in unseen target domains using only a few labeled support exemplars.

Existing CD-FSS methods (Herzog, 2024; Tong et al., 2024; Nie et al., 2024) predominantly rely on CNN backbones such as ResNet-50, and typically extend in-domain FSS either by retraining on source data with domain-generalization techniques or by mining cross-image correspondences. Although effective in controlled settings, these approaches are costly and depend on source data. Their convolutional inductive biases limit long range reasoning and robustness under distribution shift, leading to overfitting with sparse labels. Given these limitations, replacing CNN backbones with vision foundation models (VFMs) yields stronger and more transferable representations.

VFMs are pretrained vision backbones that provide transferable representations across recognition, segmentation, and multimodal tasks (Oquab et al., 2023; Chen et al., 2024c; Siméoni et al., 2025). Despite these strengths, applying them to CD-FSS is challenging for two reasons. First, the few-

Figure 1. Scarce labels and target domain shift co-occur in CD-FSS, making VFMs fragile in deployment. Retraining overfits, representations misalign, and support to query correspondence becomes unstable. Our method HERA, a three-stage select-regularize-calibrate framework adapting a frozen VFM at test time with only limited labeled exemplars and no source data, achieving state-of-the-art performance.

shot regime offers only a handful of labeled exemplars per novel class, so retraining is prone to overfitting, source dependent, and computationally prohibitive for large models. Second, distribution shifts place target domains outside pretraining coverage, so using frozen features or adapting all layers indiscriminately yields cross-domain inconsistency and pronounced layer sensitivity, which destabilize correspondence and pixel predictions. Empirically, VFM features exhibit sharp layer-wise variation in transferability under shift (Raghu et al., 2021; Lee et al., 2022; Roh et al., 2024). Adapting a single stable layer reduces degrees of freedom and curbs overfitting. Therefore, it is necessary to adopt source-free test-time adaptation (Liang et al., 2025) that first selects the most informative layer for adaptation and then keeps the remaining backbone frozen while updating only a small subset of parameters.

To address these issues, we present Hierarchical Exemplar Representation Adaptation(HERA), as shown in Fig. 1, an efficient VFM-based segmentation framework that learns from limited labels and adapts to novel domains without source data retraining. We first propose Hierarchical Layer Selection (HLS) to estimate a per episode Exemplar Transfer Risk (ETR) for each candidate layer from the few labeled supports and the forward activations, and choose the lowest risk layer as the working representation. We then confine test time updates to a small subset of parameters at this layer and keep the rest of the backbone frozen.

Even with a stable representation, token interaction maps, such as the self attention maps used in ViTs, remain noisy under distribution shift. To mitigate this, Prior Guided Regularization (PGR) regularizes the attention on the selected representation with a lightweight spatial prior, strengthening locality and structural consistency while preserving global coverage. Finally, Pixelwise Adaptive Calibration (PAC) combines the selected representation with these prior guided attention maps and a query-image prototype-contrast map to calibrate pixel-level predictions, correcting residual artifacts along thin boundaries and in low-contrast regions. In summary, our key contributions include:

- We present HERA, a source-free test-time adaptation framework for CD-FSS with VFMs that organizes adaptation as select, regularize, and calibrate, keeps the backbone frozen, and updates fewer than 2.7% of parameters at inference.

- We introduce Hierarchical Layer Selection (HLS) with a data dependent Exemplar Transfer Risk (ETR) that selects the lowest risk layer per episode from the few labeled support images and forward activations, localizing adaptation to a small parameter subset. And we further couple Prior Guided Regularization (PGR) and Pixelwise Adaptive Calibration (PAC) to regularize target-side structure and calibrate pixel-wise predictions, forming a coherent hierarchy from representation to prediction.

- Extensive experiments on multiple CD-FSS benchmarks show consistent gains over prior methods, improving by more than 6.5 mIoU, with ablations validating each stage and the layer selection criterion and confirming strong parameter efficiency.

## 2 RELATED WORK

**Few-shot semantic segmentation (FSS)** aims to predict pixel-level masks for novel classes using only a few labeled supports per class. Most existing methods fall into two families. Prototype based approaches form class prototypes in feature space and classify queries by similarity (Dong & Xing,

2018; Li et al., 2021; Lang et al., 2022; 2023). Affinity based approaches compute dense correlations or attention between support and query features to propagate context (Lu et al., 2021; Min et al., 2021; Fan et al., 2022; Peng et al., 2023). Subsequent works strengthen FSS through multi scale context aggregation, prototype refinement, and mask-level regularization (Tian et al., 2020; Peng et al., 2023; Chen et al., 2024b), yet most assume matched training and test distributions, leaving robustness to domain shift largely unexplored.

**Cross-domain few-shot segmentation (CD-FSS)** aims to generalize to novel classes in a unseen target domain using only a few labeled support exemplars. Prior work is largely CNN based and follows two lines. Source-side training performs meta learning or domain generalization on source data before deployment (Lei et al., 2022; Su et al., 2024; Chen et al., 2024b; Fan et al., 2025), which can be effective but requires continued access to source data and repeated retraining. Target domain supervised adaptation either mines correspondences across images or fine-tunes task heads and adapter modules using a few labeled supports (Wang et al., 2022b; Herzog, 2024; Tong et al., 2024; Nie et al., 2024). Despite progress, these pipelines are costly or source dependent, and convolutional inductive biases limit long range reasoning and robustness under distribution shift. In contrast, we adopt effectively VFM-based segmentation framework without source-data retraining.

**Vision foundation models (VFMs)** are largely ViT-based backbones pre-trained at scale with self, weakly, or semi supervised objectives (Oquab et al., 2023; Chen et al., 2024c; Siméoni et al., 2025; Kirillov et al., 2023). Representative models include CLIP (Radford et al., 2021) for image-text alignment, MAE and EVA02 (He et al., 2022; Fang et al., 2023; 2024) for masked-image modeling, SAM (Kirillov et al., 2023) for promptable segmentation, and DINO (Oquab et al., 2023; Siméoni et al., 2025) for self-distillation with strong objectness cues. These models provide transferable hierarchical features and often yield competitive segmentation with a frozen encoder. Specially in DINOv3 (Siméoni et al., 2025), intermediate layers present coarse-to-fine semantics and attention that is locally coherent and globally aware, which suits support–query matching. Under distribution shift, however, layer utility varies across episodes and correspondence becomes unstable, so fixed layer choices or uniform fine-tuning are unreliable. We therefore adopt ViT-based VFMs with per-episode selection of a stable layer, followed by hierarchical regularization and calibration.

**Test-time adaptation (TTA)** adapts deployed models to target data using unlabeled test samples (Wang et al., 2020; Jia et al., 2024; Liang et al., 2025). Common routes minimize entropy or consistency, update statistics, such as BN re-estimation, apply whitening or stylization, and perform contrastive or clustering-based alignment for segmentation (Wang et al., 2022a; Gong et al., 2022; Kang et al., 2024). Although deployment friendly, they optimize surrogate losses on queries, require sizable trainable subsets or lengthy per-image updates, and are weakly coupled to the episodic nature of CD-FSS. In parallel, parameter-efficient fine-tuning (PEFT) updates a small fraction of weights via adapters, prompts, or low-rank modules (Han et al., 2024; Hu et al., 2022; Xing et al., 2024; Chen et al., 2022b;a), but for dense prediction it often targets single-level proxies and lacks episode-aware alignment. Our approach unifies TTA and PEFT by updating a small parameter subset at test time on a frozen backbone, guided by stable representation selection.

## 3 INTRODUCING THE HERA FRAMEWORK

### 3.1 ARCHITECTURE OVERVIEW

Cross-domain few-shot segmentation (CD-FSS) follows an episodic $K$-shot protocol: given a support–query set $\mathcal{S} = \{(I_s^i, M_s^i)\}_{i=1}^{K}$ and a query image $I_q$, models trained on source domains are evaluated on target domains with disjoint label spaces. Leveraging vision foundation models (VFMs) is attractive. However, under distribution shift, per episode alignment between support and query becomes unstable, causing errors to cascade from representation to prediction. In addition, VFM transferability varies across layers and local interactions remain noisy.

We therefore propose Hierarchical Exemplar Representation Adaptation(HERA), a three-stage select-regularize-calibrate framework that adapts at test time with a frozen backbone. We first design Hierarchical Layer Selection (HLS) to select a stable representation layer $\ell^*$ by minimizing a data dependent Exemplar Transfer Risk (ETR) computed per episode. Then Prior Guided Regularization (PGR) refines the self attention at $\ell^*$ with entropy-gated Gaussian priors, strengthening locality and structural consistency while preserving global coverage. Finally, Pixelwise Adaptive Calibration

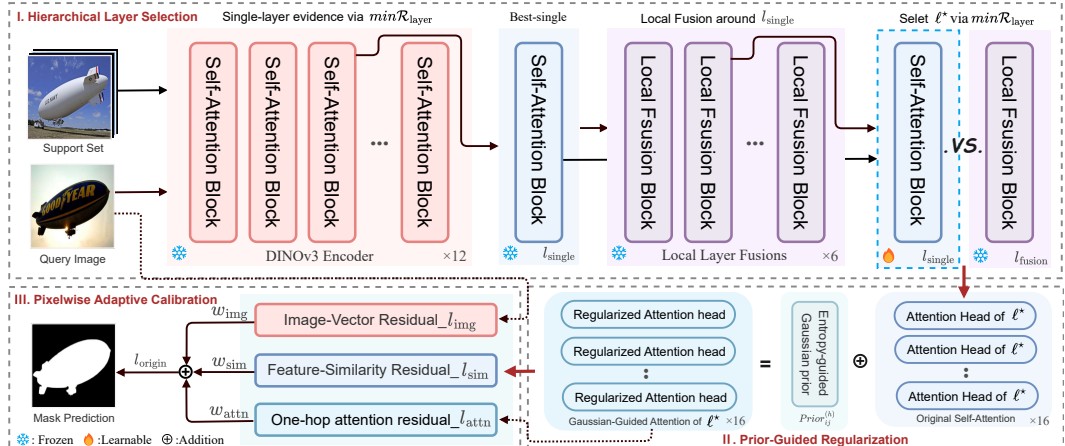

Figure 2. The proposed HERA Architecture. Hierarchical Layer Selection (HLS) estimates the leave-one-out episode risk $\mathcal{R}_{\text{layer}}$ and routes to $\ell^\star$, chosen from a single layer $\ell_{\text{single}}$ or a local-fusion set $\ell_{\text{fusion}}$. At $\ell^\star$, Prior Guided Regularization (PGR) calibrates self attention by injecting an entropy-gated Gaussian prior per head. Finally, Pixelwise Adaptive Calibration (PAC) adds residual logits $\ell_{\text{sim}}, \ell_{\text{attn}}, \ell_{\text{img}}$ and linearly fuses them with the base logit, forming a hierarchical select-regularize-calibrate pipeline.

(PAC) calibrates pixel-wise predictions via lightweight residuals to recover thin structures and denoise low-contrast regions. Together, these stages form a hierarchical path from representation to prediction and transfer a few labeled supports into reliable guidance in new domains.

## 3.2 HIERARCHICAL LAYER SELECTION

Given a frozen ViT backbone $f$ that produces layerwise features $F^\ell$, we observe pronounced cross layer transferability variance under distribution shift. We therefore make per-episode layer selection the primary stage of HERA. HLS minimizes the data dependent episode-level ETR over candidate layers and outputs the selected representation $F^{\ell^*}$, which conditions downstream modules.

### 3.2.1 LAYER WISE VARIABILITY UNDER SHIFT

As illustrated in Fig. 3, early to mid layers 0-11 emphasize low-SNR edges and textures, whereas mid to high layers 12-23 yield class-agnostic objectness with sharper boundaries, with a consistent semantic shift around layers 11-12. Because the most informative layer varies across episodes and domains, any fixed choice is brittle. Episode-wise selection is thus crucial for robust support–query alignment. We therefore restrict routing to layers 12-23, where semantics consolidate while spatial detail is preserved. Although single-layer routing in this band often performs well but fragile on thin structures, occlusions, and clutter, as layers exhibit complementary failure modes. The optimal layer may also fluctuate across episodes within the same domain. To curb this instability, we augment the single-layer choice with a compact local-fusion candidate set centered at the best single layer and evaluate all candidates under a unified episode-level risk. This adds negligible cost and yields a more reliable routed representation for subsequent modules.

### 3.2.2 PER-EPISODE LAYER SELECTION VIA EXEMPLAR TRANSFER

Following Sec. 3.2.1, we estimate episode-level evidence at test time using leave-one-out and adapt only a small subset of parameters. Let the support set be $\mathcal{S} = \{(I_s^i, M_s^i)\}_{i=1}^K$. In the $i$-th iteration, $(I_s^i, M_s^i)$ acts as the pseudo-query $S_q^i$ and the remaining supports form $S^{(-i)}$. We compute a prototype $\mathbf{P}_s^i$ from $S^{(-i)}$ and extract the pseudo-query feature $\mathbf{F}_q^i$ from $I_s^i$ at a candidate layer $\ell$.

We define Exemplar Transfer Risk (ETR) as one minus the average pseudo-query mIoU:

$$\mathcal{R}_{\text{layer}} = 1 - \frac{1}{K}\sum_{i=1}^K \text{mIoU}\Big(\cos(\mathbf{P}_s^i, \mathbf{F}_q^i), M_q^i\Big), \qquad \ell^\star = \arg\min_{\ell \in \mathcal{C}} \mathcal{R}_{\text{layer}}(\ell), \qquad (1)$$

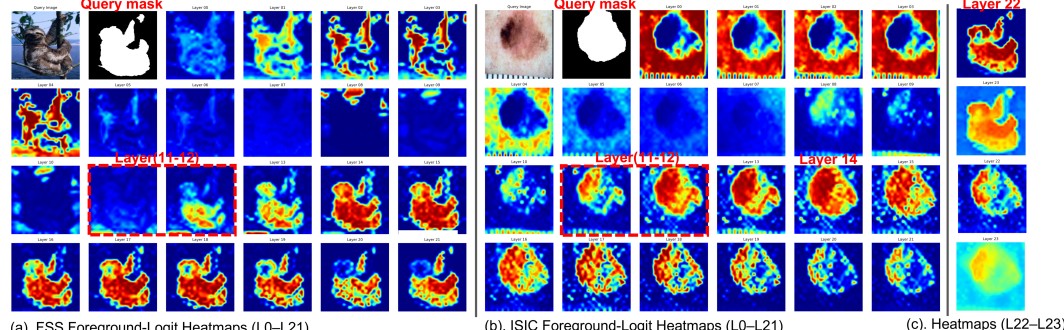

(a). FSS Foreground-Logit Heatmaps (L0–L21)    (b). ISIC Foreground-Logit Heatmaps (L0–L21)    (c). Heatmaps (L22–L23)

Figure 3. Layerwise variability in DINOv3. Layerwise foreground-logit heatmaps across ViT layers 0-23 for two episodes from distinct target domains. A clear semantic shift emerges around layers 11-12, yet the optimal layer for mask prediction differs across episodes, indicating episode and domain dependent variability in the best layer. This variability motivates routing to an episode-specific layer.

---

**Algorithm 1** Hierarchical Layer Selection (HLS)

---

1: **Require:** $K$-shot support set $\mathcal{S}$, candidate layers $\mathcal{L} = \{12, \ldots, 23\}$ and DINOv3 backbone $f$.
2: Select trainable parameters
3: **for** $n = 1$ to $K{-}1$ **do**                    ◇ *Episode-level routing evidence (n-shot): see Sec. 3.2*
4:     ▷ *Stage I: Single-layer evidence*
5:         Assemble the $n$-shot episode $S_n$; extract per-layer features $\{F^\ell\}_{\ell \in \mathcal{L}}$ with $f$
6:         Compute $\mathcal{R}_{\text{layer}}(\ell)$ on $S_n$ and obtain $\ell_{\text{single}}$            *single risk, see Eq. (1)*
7:     ▷ *Stage II: Local fusion around $\ell_{single}$*
8:         Construct a local fusion pool $\mathcal{U}(\ell_{\text{single}})$
9:         For each $U \in \mathcal{U}$, compute fusion weights $w_\ell$ and fused feature $F^U$            *see Eq. (3)*
10:        Choose $l^\star = \arg\min_{\{\ell_{\text{single}}\} \cup \mathcal{U}(\ell_{\text{single}})} \mathcal{R}_{\text{layer}}(l)$ and fix $F^{\ell^\star}$            *unified risk, see Eq. (1)*
11:    **for** $i = 1$ to $K$ **do**                    ◇ *Pseudo–query cross-evaluation*
12:        Form $\langle S^{(-i)}, (I^i_s, M^i_s) \rangle$ at fixed $F^\star$
13:        Compute $\mathcal{L}^{(i,n)}_{\text{TTA}}$ over all $n$-support combinations            *see Eq. (2)*
14:    **end for**
15:    Update $\phi$ by back-propagating the average loss $\mathcal{L}^{(n)}_{\text{TTA}}$; keep $f$ frozen        ◇ *Parameter-efficient TTA*
16: **end for**

---

where $\mathcal{C}$ contains both single-layer and local-fusion candidates, and $\text{mIoU}(\cdot)$ compares the predicted mask with the pseudo-query ground truth $M^i_q$, where $M^i_q{=}M^i_s$. The minimizer $\ell^\star$ is the routed layer used in subsequent stages.

With $\ell^\star$ fixed, we freeze the backbone and finetune only a small parameter set $\phi$ using the same leave-one-out construction, optimizing a binary segmentation loss on probabilities:

$$\mathcal{L}_{\text{TTA}} = \frac{1}{K} \sum_{i=1}^{K} \text{BCE}\Big(\cos(\mathbf{P}^{i,\ell^\star}_s, \mathbf{F}^{i,\ell^\star}_q), M^i_q\Big), \tag{2}$$

where $\mathbf{P}^{i,\ell^\star}_s$ and $\mathbf{F}^{i,\ell^\star}_q$ are computed at the routed layer $\ell^\star$. To mitigate overfitting (He et al., 2020; Boudiaf et al., 2021), we fine-tune only the selected layer's mlp.fc for single-layer routing and fusion-mlp.fc for local-fusion routing (see Sec. A) and all remaining parameters are frozen.

### 3.2.3 TWO-STAGE SELECTION AND PARAMETER-EFFICIENT ADAPTATION

Building on the analysis in Sec. 3.2.2, we first determine the best single layer $\ell_{\text{single}}$ via leave-one-out evidence, and then form a compact set of local-fusion candidates $U \subseteq \mathcal{L} = \{12, \ldots, 23\}$ in its neighborhood. Because the last layer of DINOv3 exhibits the strongest semantic aggregation in Fig. 3, every multi-layer candidate includes $\ell_{23}$ (see Sec. B) to compensate occlusion or fragmented shapes. All candidates are evaluated under the same episode-level risk ETR in Eq. (1).

For any candidate $U$, let $r_\ell = \mathcal{R}_{\text{sel}}(\ell)$ denote the single-layer selection risk. Define the distance $d(\ell, \ell_{23}) = |\ell - \ell_{23}|$ and compute fusion weights and the fused feature as:

Figure 4. Prior Guided Regularization (PGR). Per-head Gaussian priors, gated by entropy, are injected into $QK^\top$ logits to calibrate self attention at $\ell^\star$, locality is strengthened, far-field spurious peaks are suppressed, while preserving the necessary global coverage.

$$w_\ell \ = \ \frac{\exp\big(-\beta\, r_\ell - \mathrm{dist}(\ell, \ell_{23})/\tau\big)}{\sum_{j \in U} \exp\big(-\beta\, r_j - \mathrm{dist}(j, \ell_{23})/\tau\big)}, \qquad F^U \ = \ \sum_{\ell \in U} w_\ell\, F^\ell, \tag{3}$$

where $\beta > 0$ controls reliance on the evidence $r_\ell$ and $\tau > 0$ is a locality bandwidth that favors deeper, semantically aggregated layers. As $\tau \to \infty$, the locality term vanishes; as $\beta \to \infty$, the weights collapse to $\arg\min_{\ell \in U} r_\ell$, approaching the single-layer case. When evidence spreads across adjacent layers, a moderate $\tau$ trades off evidence and aggregation, reducing routing instability.

Fixing the routed layer at $\ell^\star$ mitigates the layer-wise component of episode-wise matching risk, which provides a stable, episode-conditioned representation for downstream adaptation.

### 3.3 PRIOR GUIDED REGULARIZATION

With the routed layer fixed by HLS, the representation provides stable global semantics, yet head specific self attention remains noisy under shift. Because DINO features provide mainly layer-level guidance, head-level maps show spurious long range links, insufficient coverage of nearby neighborhoods and thin boundaries, and strong cross head heterogeneity. A uniform, head-agnostic prior is therefore inadequate. We calibrate attention per head using a query-centered Gaussian prior whose bandwidth is set by an entropy gate derived from head's attention. Local and confident heads receive a sharper prior, whereas globally dispersed heads receive a more diffuse prior. This head-wise, entropy-gated calibration enforces locality while preserving necessary global coverage.

**Head-wise Gaussian Prior with Entropy Gating.** Attention heads in ViTs exhibit specialization in spatial scope and semantics (Raghu et al., 2021; Lee et al., 2022; Roh et al., 2024). We therefore impose a head-wise Gaussian prior and set its bandwidth by an entropy gate, yielding a sharper prior for local, confident heads and a more diffuse prior for globally dispersed ones:

$$\phi(p_j;\, p_i, \sigma) = \exp\Big(-\frac{\|p_j - p_i\|^2}{2\sigma^2}\Big), \tag{4}$$

with two predefined bandwidths, $\sigma_{\mathrm{loc}} < \sigma_{\mathrm{glo}}$, capturing local aggregation and global coverage. Let $\bar{H}_q^{(h)}$ denote the mean row entropy of the $QK^\top$ logits at head $h$, indicating global dispersion, and $\bar{H}_k^{(h)}$ that of $KK^\top$, indicating local stability. Using a logistic gate $g(\cdot)$ with temperature $\alpha > 0$:

$$\gamma_h = g\big(\alpha(\bar{H}_q^{(h)} - \bar{H}_k^{(h)})\big), \qquad \sigma_h = (1 - \gamma_h)\, \sigma_{\mathrm{glo}} + \gamma_h\, \sigma_{\mathrm{loc}}, \tag{5}$$

where heads with stronger locality, indicating larger $\bar{H}_q^{(h)} - \bar{H}_k^{(h)}$, receive a sharper prior, globally dispersed heads receive a more diffuse prior.

**Logit-Level Injection.** We modify only the patch-grid submatrix for head $h$; CLS and register tokens remain unchanged in DINOv3. Let $L_{\mathrm{pp}}^{(h)}$ denote the $QK^\top$ logits block and the calibrated

attention is obtained by adding the log-prior to the logits:

$$P_{ij}^{(h)} = \frac{\phi(p_j; \, p_i, \sigma_h)}{\sum_{j'} \phi(p_{j'}; \, p_i, \sigma_h)}, \quad \widetilde{A}_{\mathrm{pp}}^{(h)} = \mathrm{softmax}\Big( L_{\mathrm{pp}}^{(h)} + \log P^{(h)} \Big). \quad (6)$$

This per-head, entropy-gated calibration enforces data adaptive locality, tightening near-field focus, damping far-field artifacts, and preparing the attention for subsequent pixelwise refinement.

### 3.4 Pixelwise Adaptive Calibration

With the layer and attention stages in place, pixelwise decisions remain unstable under shift, leaving residual errors along thin boundaries and in low-contrast regions. PAC explicitly performs pixelwise calibration by fusing three lightweight cues computed from $F^{\ell^*}$ and the refined attention, namely feature similarity, one hop attention propagation, and image appearance. This fusion corrects residual artifacts and yields consistent masks:

$$\ell_{\mathrm{final}}(x) = \ell_0(x) + w_{\mathrm{sim}}\ell_{\mathrm{sim}}(x) + w_{\mathrm{attn}}\ell_{\mathrm{attn}}(x) + w_{\mathrm{img}}\ell_{\mathrm{img}}(x), \quad (7)$$

where $\ell_0(x)$ is the base logit from the selected representation and $w.$ are fixed scalar weights (see Sec. C ). A single-step refine-vote gate applies residuals only when the estimated gain is positive, adding negligible overhead (see Sec. D). Taken together, the three stages realize a hierarchical select-regularize-calibrate pipeline that adapts at test time with a frozen backbone.

## 4 EXPERIMENTS

**Evaluation Datasets and Metrics.** We evaluate in a source-free test-time adaptation setting without access to source data. And we evaluate on four public target datasets: FSS-1000 (Li et al., 2020), a natural image corpus spanning 1,000 object classes; DeepGlobe (Demir et al., 2018), a satellite land cover dataset with seven categories and pronounced texture and scale shifts; ISIC2018 (Tschandl et al., 2018; Codella et al., 2019), a dermoscopic skin lesion dataset with low-contrast and irregular boundaries; and Chest X-ray (Jaeger et al., 2013; Candemir et al., 2013), a medical radiograph dataset with substantial grayscale and structural variation. We follow the episodic K-shot protocol and report mean IoU for 1-shot and 5-shot. All methods use identical episode sampling, a shared preprocessing pipeline, and a unified input resolution of $400 \times 400$.

**Implementation Details.** We use SSP (Fan et al., 2022) as the few-shot head and run HERA on a ViT backbone (default DINOv3 (Siméoni et al., 2025)). Under test-time adaptation (TTA), each target episode proceeds as follows: (i) HLS selects the routed layer, and (ii) we form leave-one-out splits of the $K$-support examples and minimize the loss in equation 2 on these splits, performing $K-1$ lightweight updates. Only a small parameter subset is trainable: the single-layer variant updates that layer's mlp.fc, and the local-fusion variant updates fusion-mlp.fc. All other weights remain frozen. We use Adam with a learning rate of $1.3 \times 10^{-3}$. In the 1-shot setting, we synthesize two augmented views of the support for TTA. All experiments run on a single NVIDIA A100. Fewer than 2.7% of parameters are updated, so compute and memory overheads are negligible.

### 4.1 Comparison Experiments

In Table 1, we compare HERA with existing cross-domain few-shot segmentation (CD-FSS) methods. HERA with DINOv3 achieves 68.3/77.9 mIoU in the 1-shot and 5-shot settings, outperforming IFA (Nie et al., 2024) by 0.5/6.5 points and the no-retraining baseline SSP by 11.0/14.8 points. With a smaller backbone, HERA with DINOv2 (Oquab et al., 2023) reaches 62.6/73.4 mIoU and still surpasses most retraining-based CD-FSS approaches, indicating that explicit layer selection, attention regularization, and pixel-level calibration adapt VFMs without retraining on source data.

On DeepGlobe, where clutter and fragmented foregrounds weaken few shot cues, HERA delivers large improvements in the 5-shot setting. On ISIC and Chest X-ray, which have clean backgrounds but thin and low contrast boundaries, gains are steady, for example 11.8/13.3 mIoU over IFA on Chest X-ray in 1-shot and 5-shot. On FSS-1000, despite substantial appearance diversity across one thousand classes, the same hierarchical procedure supports robust class agnostic matching and outperforms recent retraining methods while using no source data.

Qualitative results in Fig. 5 show that HERA with a frozen ViT produces cleaner masks, reduced background leakage, sharper boundaries, and more complete object coverage.

Table 1. Quantitative comparison on the CD-FSS benchmark. All compared methods, except HERA, are trained on the Pascal VOC source domain and evaluated on four distinct targets. Best and second-best results are shown in **bold** and underline. The [†] indicates results reproduced by us. The [‡] indicates a ViT-base backbone. In-domain FSS methods are applied to CD-FSS without retraining (✗), while most CD-FSS methods retrain on the source domain (✓) to enhance generalization.

| Methods | Publication | Retraining | DeepGlobe | | ISIC | | Chest X-ray | | FSS-1000 | | mIoU | |
|---|---|---|---|---|---|---|---|---|---|---|---|---|
| | | | 1-shot | 5-shot | 1-shot | 5-shot | 1-shot | 5-shot | 1-shot | 5-shot | 1-shot | 5-shot |
| PGNet (Zhang et al., 2019a) | ICCV 2019 | ✗ | 10.7 | 12.4 | 21.9 | 21.3 | 34.0 | 23.0 | 62.4 | 62.7 | 32.2 | 31.1 |
| PANet (Wang et al., 2019) | ICCV 2019 | ✗ | 36.6 | 45.4 | 25.3 | 34.0 | 57.8 | 69.3 | 69.2 | 71.7 | 47.2 | 55.1 |
| CaNet (Zhang et al., 2019b) | CVPR 2019 | ✗ | 22.3 | 23.1 | 25.2 | 28.2 | 28.4 | 28.6 | 70.7 | 72.0 | 36.6 | 38.0 |
| RPMMs (Yang et al., 2020) | ECCV 2020 | ✗ | 13.0 | 13.5 | 18.0 | 20.0 | 30.1 | 30.8 | 65.1 | 67.1 | 31.6 | 32.9 |
| PFENet (Tian et al., 2020) | TPAMI 2020 | ✗ | 16.9 | 18.0 | 23.5 | 23.8 | 27.2 | 27.6 | 70.9 | 70.5 | 34.6 | 35.0 |
| RePRI (Boudiaf et al., 2021) | CVPR 2021 | ✗ | 25.0 | 27.4 | 23.3 | 26.2 | 65.1 | 65.5 | 71.0 | 74.2 | 46.1 | 48.3 |
| HSNet (Min et al., 2021) | ICCV 2021 | ✗ | 29.7 | 35.1 | 31.2 | 35.1 | 51.9 | 54.4 | 77.5 | 81.0 | 47.6 | 51.4 |
| SSP[†] (Fan et al., 2022) | ECCV 2022 | ✗ | 40.5 | 49.6 | 35.5 | 48.2 | 74.2 | 74.5 | 79.0 | 80.2 | 57.3 | 63.1 |
| PATNet (Lei et al., 2022) | ECCV 2022 | ✓ | 37.9 | 43.0 | 41.2 | 53.6 | 66.6 | 70.2 | 78.6 | 81.2 | 56.1 | 62.0 |
| PMNet (Chen et al., 2024a) | WACV 2024 | ✓ | 37.1 | 41.6 | 51.2 | 54.5 | 70.4 | 74.0 | **84.6** | 86.3 | 60.8 | 64.1 |
| ABCDFSS (Herzog, 2024) | CVPR 2024 | ✓ | 42.6 | 49.0 | 45.7 | 53.3 | 79.8 | 81.4 | 74.6 | 76.2 | 60.7 | 65.0 |
| APSeg[‡] (He et al., 2024) | CVPR 2024 | ✓ | 35.9 | 40.0 | 45.4 | 54.0 | 84.1 | 84.5 | 79.7 | 81.9 | 61.3 | 65.1 |
| DR-Adapter (Su et al., 2024) | CVPR 2024 | ✓ | 41.3 | 50.1 | 40.8 | 48.9 | 82.4 | 82.3 | 79.1 | 80.4 | 60.9 | 65.4 |
| APM (Tong et al., 2024) | NeurIPS 2024 | ✓ | 40.9 | 44.9 | 41.7 | 51.2 | 78.3 | 82.8 | 79.3 | 81.9 | 60.0 | 65.2 |
| IFA (Nie et al., 2024) | CVPR 2024 | ✓ | **50.6** | 58.8 | **66.3** | 69.8 | 74.0 | 74.6 | 80.1 | 82.4 | 67.8 | 71.4 |
| LoEC[‡] (Liu et al., 2025) | CVPR 2025 | ✓ | 42.1 | 51.5 | 52.9 | 62.4 | 83.9 | 84.1 | 81.1 | 83.7 | 65.0 | 70.4 |
| HERA[‡] (DINOv2) | – | ✗ | 41.2 | 57.8 | 55.6 | 68.7 | 83.2 | 86.9 | 70.2 | 80.3 | 62.6 | 73.4 |
| HERA[‡] (DINOv3) | – | ✗ | 44.6 | **63.4** | 61.2 | **73.6** | **85.8** | **87.9** | 81.6 | **86.7** | **68.3** | **77.9** |

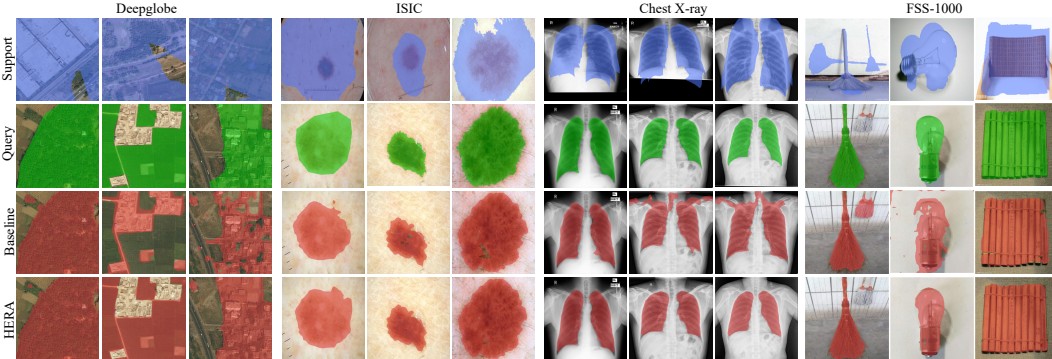

Figure 5. Qualitative results on the Chest X-ray, ISIC, FSS-1000, and Deepglobe datasets under the 1-shot setting. The prediction and ground truth of query images are in red and green, respectively.

Table 2. Ablation studies for components of our method.

| Method | DeepGlobe | ISIC | Chest X-ray | FSS-1000 | 5-shot | Δ Avg. |
|---|---|---|---|---|---|---|
| Baseline | 49.6 | 48.2 | 74.5 | 80.2 | 63.1 | 0.0 |
| + HLS | 61.7 | 71.4 | 87.7 | 86.0 | 76.7 | +13.6 |
| + HLS + PGR | 62.6 | 72.0 | 88.0 | 86.5 | 77.3 | +14.2 |
| + HLS + PAC | 62.1 | 71.6 | **88.3** | 86.6 | 77.2 | +14.1 |
| **+ HLS + PGR + PAC** | **63.4** | **73.6** | 87.9 | **86.7** | **77.9** | **+14.8** |

Table 3. PAC branch ablation.

| Variant | 5-shot | Δ Avg. |
|---|---|---|
| Baseline + HLS + PGR | 77.27 | – |
| + l-sim | 77.57 | +0.30 |
| + l-attn | 77.49 | +0.22 |
| + l-img | 77.45 | +0.18 |
| **+ l-sim + l-attn + l-img** | **77.91** | **+0.64** |

## 4.2 ABLATION STUDIES

**Component ablation.** We ablate HERA in the 5-shot setting with a frozen VFM backbone as shown in Table 2. The SSP baseline averages 63.1 mIoU. Adding HLS lifts the mean to 76.7 mIoU, a gain of 13.6 mIoU and the major source of improvement. Building on HLS, PGR and PAC provide complementary refinements, reaching 77.3 mIoU and 77.2 mIoU, corresponding to gains of 14.2 mIoU and 14.1 mIoU over the baseline. The full stack attains 77.9 mIoU, a total gain of 14.8 mIoU. The largest per-dataset improvement appears on ISIC, from 48.2 to 73.6 mIoU, a gain of 25.4 mIoU. Overall, per-episode layer selection accounts for most of the benefit, and attention regularization and pixel-level calibration add stable complementary gains, consistent with the top-down select–regularize–calibrate design.

Table 4. Layer selection ablation with a frozen backbone (5-shot mIoU↑). Rule lists the per episode selector and notation includes $\ell$ layer index; $\mathbf{g}_\ell$ gradient w.r.t. features of layer $\ell$; $S_{\text{sem}}$, $S_{\text{str}}$, $C$ normalized semantic, structure, and complexity scores; and $\text{mIoU}_{\text{sup}}(\ell)$ support only mIoU.

| Method | Rule | DeepGlobe | ISIC | Chest X-ray | FSS-1000 | Avg. | Δ |
|---|---|---|---|---|---|---|---|
| Static-Max | $\arg\max_\ell \left( \alpha\, S_{\text{sem}}(\ell) + \beta\, S_{\text{str}}(\ell) + \gamma\, C(\ell) \right)$ | 58.8 | 67.2 | 80.1 | 81.4 | 71.9 | 0.0 |
| Grad-Max | $\arg\max_\ell \|\mathbf{g}_\ell\|$ | 60.2 | 62.0 | 85.4 | 84.7 | 73.1 | +1.2 |
| GradΔ-Max | $\arg\max_\ell \|\mathbf{g}_\ell - \mathbf{g}_{\ell-1}\|$ | 60.4 | 61.7 | 86.0 | 84.8 | 73.2 | +1.3 |
| **HLS (ETR)** | $\arg\min_\ell \left( 1 - \text{mIoU}_{\text{sup}}(\ell) \right)$ | **61.7** | **71.4** | **87.7** | **86.0** | **76.7** | **+4.8** |

**PGR-PAC interplay and PAC branches.** On top of HLS at 76.7 mIoU, adding PGR lifts the mean to 77.3 mIoU, adding PAC yields 77.2 mIoU, and enabling both reaches 77.9 mIoU; the corresponding gains over HLS are 0.6, 0.5, and 1.2 mIoU, see Table 2. Starting from HLS + PGR at 77.27 mIoU, Table 3 further decomposes PAC: the similarity residual $\ell_{\text{sim}}$, the one hop attention prior $\ell_{\text{attn}}$, and the image cue $\ell_{\text{img}}$ add 0.30, 0.22, and 0.18 mIoU, respectively, and using all three reaches 77.91 mIoU, an additional 0.64 mIoU over HLS + PGR. Thus, with a frozen backbone, layer selection, attention regularization, and pixel-level calibration act at complementary levels and together yield cumulative gains.

**Layer selection rationale.** We compare per-episode selectors for test-time adaptation with a frozen backbone, as shown in Table 4. Our criterion based HLS routes each episode to the layer that maximizes support only mIoU, $\arg\min_\ell \left( 1 - \text{mIoU}_{\text{sup}}(\ell) \right)$, and provides task aligned, episode aware feedback without extra parameters or surrogate losses. In contrast, Static Max scores feature quality using semantic, structural, and complexity heuristics rather than task fit. Gradient proxies, including Grad Max and GradΔ Max, rank layers by gradient magnitude or change. These proxies tend to favor the final blocks in ViTs because of residual connections and normalization, and they correlate weakly with semantic alignment. HLS attains the best average of 76.7 mIoU. This is 4.8 above Static Max and 3.5 above the strongest proxy, which reaches 73.2. On ISIC the gain is large, from 48.2 to 73.6 mIoU, an improvement of 25.4. These results highlight per-episode layer selection as the primary driver and justify using HLS as the entry point of HERA (see Sec. E). The gap is larger on other VFM backbones, for example DINOv2 (Oquab et al., 2023).

**Deployment cost in a new target domain.** Many retraining and domain generalization methods require tens to hundreds of GPU hours on source data prior to deployment. Our method requires no source-data retraining and directly leverages VFMs. For a new target domain, we perform a single adaptation step using the available supports. In the 1 shot setting, this adaptation takes 0.735 s, including 0.202 s for layer routing and 0.280 s for a lightweight parameter update. Only 2.69% of parameters are trainable during this step. After adaptation, the same model is used for other images from that domain with no further training. These measurements show that HERA enables source-free and parameter-efficient adaptation for CD-FSS, providing high generalization efficiency, convenient deployment, and low compute cost.

## 5 CONCLUSION

We first identify the primary bottleneck in applying vision foundation models (VFMs) to cross-domain few-shot segmentation (CD-FSS) as layer-wise transferability variation, together with noisy head-level interactions under shift, rather than limited representational capacity. We propose HERA, a source-free test-time adaptation framework that turns a few labeled supports into reliable guidance for VFMs in CD-FSS. HERA keeps the ViT backbone frozen and performs a three-stage select-regularize-calibrate procedure. We design Hierarchical Layer Selection (HLS) to identify a stable representation using the data-dependent Exemplar Transfer Risk (ETR) over candidate layers, then Prior Guided Regularization (PGR) regularizes per-head attention on the selected representation using an entropy-gated Gaussian prior that strengthens locality while preserving global coverage, and Pixelwise Adaptive Calibration (PAC) finally fuses complementary signals, feature prototype similarity, one-hop attention propagation, and image appearance to calibrate pixel-level logits, correcting thin-boundary and low-contrast errors. Empirically, HERA surpasses the state-of-the-art by more than 6.5 mIoU with low overhead and practical deployability across domains and backbones, providing a lightweight recipe for leveraging VFMs in CD-FSS.

## ETHICS STATEMENT

All authors have read and agree to the ICLR Code of Ethics. This work involves no interventions with human participants and collects no personally identifiable information. We use only publicly available datasets under their original licenses and follow the stated terms of use. Potential risks and mitigations are summarized below.

**Privacy and security.** No personal data are collected or released. Medical and dermoscopic images come from public, de identified sources, and any residual metadata are not used. Qualitative examples are drawn exclusively from public datasets.

**Bias and fairness.** Benchmarks may contain demographic, geographic, or acquisition biases. We report results across multiple domains, provide complete configuration details for external auditing, and encourage further evaluation on broader populations and sensing conditions.

**Dual use and misuse.** The method could be repurposed for large scale monitoring or clinical triage without proper oversight. Our artifacts are for research only. We do not release web scraping, re identification, or deployment tools, and the models are not intended for clinical decision making.

**Legal compliance.** We comply with the licenses of all third party assets (code, models, datasets) and cite their sources. Any additional third party terms are respected.

**Research integrity and environmental impact.** We document preprocessing, adaptation protocols, and hyperparameters to support reproducibility. Parameter efficient test time adaptation reduces compute for training relative to end to end retraining. We report hardware and runtime to facilitate cost estimation. Where applicable, institutional review details are withheld for double blind review and can be provided upon acceptance.

## REPRODUCIBILITY STATEMENT

We provide the details required to reproduce our results: (i) complete hyperparameters, optimizer settings, and adaptation budgets; (ii) dataset preprocessing, links, and splits, with episode sampling policies that are seeded, one shot and five shot; (iii) code structure with scripts to reproduce all main tables and figures, including ablations of HLS, PGR, and PAC, and the layer selection criterion; (iv) checkpoints, logs, and exact trainable parameter counts; (v) hardware specifications (single NVIDIA A100), input resolution ($400 \times 400$), and per episode runtime. All dependencies are version pinned, with deterministic flags and seeds provided to enable bitwise stable reruns.

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

## A  MLP ADAPTATION ON THE SELECTED REPRESENTATION LAYER

**MLP head variants.**  With the backbone frozen and the working representation layer chosen by HLS, we attach a small MLP head at that layer to improve support to query correspondence at test time. We evaluate three variants: M0, no MLP head (apply_fc=False); M1, MLP branch present but frozen (apply_fc=True, zero_init=True; parameters fixed); and M2, a trainable MLP fine tuned at test time on the selected layer (apply_fc=True). Only the MLP head is updated, keeping the fraction of updated parameters below 2.7%.

Table A.1.  MLP ablation at the layer selected by HLS with the backbone frozen.  $\Delta$ denotes the improvement relative to the row above.

| Variant | mIoU@1 | $\Delta$ | mIoU@5 | $\Delta$ |
|---------|--------|----------|--------|----------|
| M0: no MLP | 65.66 | – | 75.20 | – |
| M1: MLP frozen | 66.33 | +0.67 | 75.78 | +0.58 |
| M2: trainable MLP | **68.29** | **+1.96** | **77.91** | **+2.13** |

**Analysis.**  Starting from M0 at 65.66 mIoU in one shot and 75.20 mIoU in five shot, as shown in Table A, adding a frozen residual MLP branch (M1) raises the means to 66.33 and 75.78 mIoU, with gains of 0.67 and 0.58 over M0. This suggests that even a fixed projection stabilizes channel scales and token mixing at the selected layer. Allowing this compact head to adapt at test time (M2) further increases accuracy to 68.29 and 77.91 mIoU, adding 1.96 and 2.13 over M1. Cumulatively, M2 improves over M0 by 2.63 in one shot and 2.71 in five shot, which correspond to relative gains of about 4.0% and 3.6%, while keeping the fraction of updated parameters under 2.7%. These gains are consistent with the Select Regularize Calibrate design. HLS provides a stable representation. The small MLP recenters and rescales features to reduce support to query mismatch, and the resulting representations interact more reliably with PGR and PAC. In practice, a single compact trainable MLP on the selected layer delivers most of the benefit with minimal overhead.

## B  LOCAL FUSION AROUND THE ROUTED LAYER

After HLR selects the best single layer $\ell_{\text{single}}$ for each episode we form a compact neighborhood $U$ centered at $\ell_{\text{single}}$ and we include the last ViT layer $L_{23}$ to mitigate fragmented shapes. We evaluate all candidates under the same episodic objective as in Sec. 3.2.1. For any $U$ let $r_\ell$ denote the single layer ETR of layer $\ell$. We compute the fusion weights and the fused representation as follows:

$$w_\ell = \frac{\exp\left(-\beta\, r_\ell - \text{dist}(\ell, \ell_{23})/\tau\right)}{\sum_{j \in U} \exp\left(-\beta\, r_j - \text{dist}(j, \ell_{23})/\tau\right)}, \qquad F^U = \sum_{\ell \in U} w_\ell\, F^\ell, \qquad \text{(B.1)}$$

Here $\beta > 0$ controls reliance on the data evidence $r_\ell$, and $\tau > 0$ is a locality bandwidth that biases the fusion toward deeper semantically aggregated layers. As $\tau \to \infty$ the locality term vanishes and the solution reduces to single layer routing, that is $\arg\min_{\ell \in U} r_\ell$. When evidence spreads across adjacent layers a moderate $\tau$ balances data evidence and semantic aggregation and stabilizes routing.

Table B.1. Local fusion centered at the routed layer. We report average mIoU for the one shot and five shot settings, along with the changes relative to using $L_{23}$ alone and to excluding $L_{23}$.

| Variant | mIoU avg. | | $\Delta$ vs. $L_{23}$ | $\Delta$ vs. no $L_{23}$ |
|---------|-----------|-----------|-----------------------|--------------------------|
| | 1 shot | 5 shot | 5 shot | 5 shot |
| $F^0 + L_{23}$, $\tau{=}0.0$ | 66.58 | 75.49 | 0.00 | 0.00 |
| $F^0 + L_{23}$, $\tau{=}2.0$ | **68.29** | **77.85** | **2.36** | **2.36** |
| $F^0 +$ no $L_{23}$, pivot=last, $\tau{=}0.0$ | 66.45 | 75.29 | $-0.20$ | 0.00 |
| $F^0 +$ no $L_{23}$, pivot=$\ell^\star$, $\tau{=}2.0$ | 66.83 | 76.34 | 0.85 | 1.05 |

**Analysis.**  Table B.1 compares single layer routing with local fusion. Local fusion centered at $L_{23}$ with $\tau{=}2.0$ outperforms using $L_{23}$ alone on both one shot and five shot averages. Excluding $L_{23}$ from the candidate set reduces performance. Redirecting fusion to the routed layer $\ell^\star$ recovers part of the performance drop, yet it remains inferior to configurations that include $L_{23}$. By dataset,

Table B.2. By dataset mIoU comparing $L_{23}$ alone and local fusion. Including $L_{23}$ in the candidate pool and setting $\tau{=}2.0$ yields the highest averages, with the largest gains on DeepGlobe and ISIC.

| Backbone (DINOv3) | DeepGlobe | | ISIC | | Chest X-ray | | FSS-1000 | | Average | |
|---|---|---|---|---|---|---|---|---|---|---|
| | 1 shot | 5 shot | 1 shot | 5 shot | 1 shot | 5 shot | 1 shot | 5 shot | 1 shot | 5 shot |
| $F^0 + L_{23}, \tau{=}2.0$ | **44.59** | **63.43** | **61.17** | **73.64** | 85.80 | 87.88 | **81.59** | **86.69** | **68.29** | **77.91** |
| $F^0 + L_{23}, \tau{=}0.0$ | 42.90 | 61.49 | 55.17 | 66.53 | 87.06 | 88.29 | 81.20 | 85.63 | 66.58 | 75.49 |
| $F^0 + $ no $L_{23}$, pivot=last, $\tau{=}0.0$ | 42.87 | 61.43 | 54.84 | 66.00 | 87.01 | 88.26 | 81.09 | 85.47 | 66.45 | 75.29 |
| $F^0 + $ no $L_{23}$, pivot=$\ell^\star$, $\tau{=}2.0$ | 42.32 | 63.11 | 56.41 | 68.25 | **87.44** | **88.41** | 81.16 | 85.58 | 66.83 | 76.34 |

Table B.2 reports larger gains on DeepGlobe and ISIC, consistent with evidence drift across episodes and the need for deeper semantic aggregation. Therefore, we adopt local fusion with $\tau{=}2.0$ and retain $L_{23}$ in the candidate pool by default.

## C    PIXELWISE ADAPTIVE CALIBRATION: DETAILS

Despite HLS and PGR, residual errors persist along thin boundaries, slender structures, and low contrast regions. With the backbone frozen, PAC adds three lightweight residual branches in the logit domain, coupled to the routed layer $\ell^*$ and to the patch attention calibrated by PGR.

**Feature similarity for semantic alignment.**    Let $\mathbf{F}_q(x)$ denote the query feature at $\ell^*$. Foreground and background prototypes, $\mathbf{P}_{fg}$ and $\mathbf{P}_{bg}$, are computed by masked averaging over support features at $\ell^*$. We define the prototype difference logit as

$$\ell_{sim}(x) = \tau_{sim}\big[\cos\big(\mathbf{F}_q(x), \mathbf{P}_{fg}\big) - \cos\big(\mathbf{F}_q(x), \mathbf{P}_{bg}\big)\big], \tag{C.1}$$

where $\tau_{sim}$ is a small temperature. This branch recovers missed regions and sharpens local focus.

**One hop attention for spatial consistency.**    Let $\widetilde{A}$ denote the row normalized patch to patch attention at $\ell^*$ after PGR. Given the base foreground probability $p_0(x) = \sigma(\ell_0(x))$, we propagate once on the patch grid as

$$\ell_{attn}(x) = \tau_{attn}\big[(\widetilde{A}\,p_0)_x\big], \tag{C.2}$$

This elongates responses along the object extent and suppresses spurious long range peaks, with limited impact on the global distribution.

**Image vector for appearance correction.**    Let $\mathbf{v}(x)$ denote a shallow appearance embedding for color and texture as

$$\ell_{img}(x) = \tau_{img}\big[\cos\big(\mathbf{v}(x), \mathbf{u}_{fg}\big) - \cos\big(\mathbf{v}(x), \mathbf{u}_{bg}\big)\big], \tag{C.3}$$

Here $\mathbf{u}_{fg}$ and $\mathbf{u}_{bg}$ are image level prototypes, and $\tau_{img}$ is a small temperature. This branch provides light global denoising and prevents over shrinking.

**The final logit** is a linear combination in the logit domain:

$$\ell_{final}(x) = \ell_0(x) + w_{sim}\,\ell_{sim}(x) + w_{attn}\,\ell_{attn}(x) + w_{img}\,\ell_{img}(x), \tag{C.4}$$

where $\ell_0(x)$ is the base logit from the selected representation and $w_.$ are fixed scalar weights. A single step refine vote gate applies residuals only when the estimated gain is positive, adding negligible overhead. Together, the three stages realize a hierarchical Select, Regularize, and Calibrate pipeline that adapts at test time with a frozen backbone.

## D    ADAPTIVE GATING FOR PIXELWISE ADAPTIVE CALIBRATION

After HLS and PGR, residual errors concentrate along thin boundaries and in low contrast regions. Pixelwise Adaptive Calibration (PAC) adds three lightweight residual branches in the logit domain, namely feature similarity, one hop attention propagation, and image appearance, while the backbone remains frozen.

To avoid negative transfer, we enable PAC only when leave one out voting on the supports predicts a positive gain. Concretely, we treat each support as a pseudo query, compute the $\Delta$mIoU with and without PAC, and enable PAC on the true query if at least $T$ votes are positive. In the one shot case, we synthesize two augmented views of the support to obtain two votes.

Table D.1. Effect of PAC gating thresholds. We report average mIoU (%) and the trigger rate of the automatic gate. The best policy is to keep the gate always on for one shot, and to use automatic gating with threshold $2/5$ for five shot.

| Policy | 1 shot | 5 shot | Trigger rate (auto) |
|---|---|---|---|
| refine = off | 67.54 | 76.67 | - |
| auto, $T{=}1$ | 68.02 | - | 56.32 |
| auto, $T{=}2$ | - | **77.91** | **74.57** |
| auto, $T{=}3$ | - | 77.22 | 59.44 |
| always on | **68.29** | 77.80 | - |

Table D.2. By dataset mIoU and gate trigger rates. The recommended setting (one shot always on, five shot automatic gating with threshold $2/5$) yields the highest average mIoU.

| | DeepGlobe | | ISIC | | Chest X-ray | | FSS-1000 | | Average | | Avg. trigger rate (%) | |
|---|---|---|---|---|---|---|---|---|---|---|---|---|
| Setting | 1 shot | 5 shot | 1 shot | 5 shot | 1 shot | 5 shot | 1 shot | 5 shot | 1 shot | 5 shot | 1 shot | 5 shot |
| 1 shot auto, 5 shot always | 44.35 | 63.51 | 60.28 | 73.72 | 86.27 | 87.22 | 81.19 | 86.73 | 68.02 | 77.80 | – | – |
| Trigger rate (%) | **55.83** | – | **50.00** | – | **19.50** | – | **99.95** | – | – | – | **56.32** | – |
| 1 shot always, 5 shot auto 2/5 | 44.59 | 63.43 | 61.17 | 73.64 | 85.80 | 87.88 | 81.59 | 86.69 | **68.29** | **77.91** | – | – |
| Trigger rate (%) | – | **25.67** | – | **97.83** | – | **25.67** | – | **84.46** | – | – | – | **74.57** |
| 1 shot always, 5 shot auto 3/5 | 44.59 | 63.41 | 61.17 | 73.40 | 85.80 | 87.95 | 81.59 | 86.63 | **68.29** | 77.85 | – | – |
| Trigger rate (%) | – | 5.83 | – | 69.00 | – | 5.83 | – | 74.92 | – | – | – | 59.44 |

**Analysis.** Relative to HLS at 76.7 mIoU, PGR raises the mean to 77.3 (+0.6), PAC to 77.2 (+0.5), and using PGR together with PAC yields 77.9 (+1.2), confirming complementarity (see Table 2). For PAC gating, Table D.1 shows that in the one shot setting the best policy is to keep PAC on for all episodes (68.29 mIoU). In the five shot setting, the automatic gate with threshold $T{=}2$ out of 5 achieves the highest mean mIoU (77.91) with a moderate trigger rate (74.6%), whereas $T{=}3$ out of 5 reduces the trigger rate and lowers accuracy to 77.22 to 77.85 mIoU. The per dataset study in Table D.2 supports the same recommendation: one shot with PAC on for all episodes and five shot with automatic gating at $T{=}2$ out of 5.

Decomposing PAC on top of HLS plus PGR at 77.27 mIoU, the similarity residual $\ell_{\mathrm{sim}}$, the one hop attention propagation $\ell_{\mathrm{attn}}$, and the image appearance cue $\ell_{\mathrm{img}}$ contribute +0.30, +0.22, and +0.18 mIoU. Using all three reaches 77.91 mIoU, a further +0.64 (see Table 3). Together, HLS stabilizes the routed layer, PGR sharpens locality, and PAC corrects pixel level logits, yielding a cumulative gain under a frozen backbone.

# E ALTERNATIVE LAYER SELECTION CRITERIA AND DINOv2 RESULTS

## E.1 EPISODE NOTATION AND SETTING

Let $\ell \in \mathcal{C}$ index a ViT layer, and let $\mathbf{F}_q^\ell(x) \in \mathbb{R}^{d_\ell}$ denote the query feature at pixel $x$ from layer $\ell$. Support features are pooled using masks to form foreground and background prototypes $\mathbf{P}_{\mathrm{fg}}^\ell$ and $\mathbf{P}_{\mathrm{bg}}^\ell$. Given a baseline foreground probability $p_0(x) \in [0, 1]$ for the query, we build soft masked query prototypes as

$$\mathbf{Q}_{\mathrm{fg}}^\ell = \frac{\sum_x p_0(x)\,\mathbf{F}_q^\ell(x)}{\sum_x p_0(x)}, \qquad \mathbf{Q}_{\mathrm{bg}}^\ell = \frac{\sum_x (1 - p_0(x))\,\mathbf{F}_q^\ell(x)}{\sum_x (1 - p_0(x))}. \tag{E.1}$$

Unless noted otherwise, all scalar layer scores are range normalized *within each episode* across $\mathcal{C}$, so different selectors are comparable:

$$\tilde{s}_\ell = \frac{s_\ell - \min_{j \in \mathcal{C}} s_j}{\max_{j \in \mathcal{C}} s_j - \min_{j \in \mathcal{C}} s_j + \varepsilon}, \quad \varepsilon = 10^{-8}. \tag{E.2}$$

## E.2 SELECTORS OTHER THAN HLS: DEFINITIONS, INTUITION, AND CAVEATS

We group the non episodic selectors in Table 4 into two families: a heuristic static rule built from prototype and mask scores, and gradient based proxies. Unless noted, *all scalar layer scores are range normalized across the candidate set $\mathcal{C}$ within each episode*. Prototypes and the baseline mask $p_0$ follow the definitions in Sec. E.

Table E.1. Notation for layer selection in the episodic setting. All scalar layer scores are range normalized across the candidate set $\mathcal{C}$ unless noted.

| Symbol | Description |
|---|---|
| $\ell \in \mathcal{C}$ | Candidate ViT layer index |
| $\mathbf{F}_q^\ell(x) \in \mathbb{R}^{d_\ell}$ | Query feature at pixel $x$ from layer $\ell$ |
| $\mathbf{P}_{\text{fg}}^\ell$, $\mathbf{P}_{\text{bg}}^\ell$ | Support foreground and background prototypes at layer $\ell$ |
| $\mathbf{Q}_{\text{fg}}^\ell$, $\mathbf{Q}_{\text{bg}}^\ell$ | Soft masked query prototypes (see Eq. equation E.1) |
| $p_0(x) \in [0, 1]$ | Baseline foreground probability on the query |
| $\text{mIoU}_{\text{sup}}(\ell)$ | Support only pseudo query mIoU at layer $\ell$ (risk proxy) |

**Static heuristic selector (Static-Max).** This rule blends three normalized scores, namely semantic agreement, structure separation, and a complexity term combining texture and uncertainty, and selects the layer with the largest weighted sum:

$$\ell_{\text{static}}^\star = \arg\max_{\ell \in \mathcal{C}} \left[ \alpha' S_{\text{sem}}(\ell) + \beta' S_{\text{str}}(\ell) + \gamma' C(\ell) \right], \qquad \alpha', \beta', \gamma' \geq 0, \ \alpha' + \beta' + \gamma' = 1. \quad \text{(E.3)}$$

*Caveat*: weights are domain and task specific, and the objective is a surrogate not directly tied to episode level mIoU risk.

**Component scores of Static-Max.**

- **Semantic agreement**

$$S_{\text{sem}}(\ell) = \alpha \cos\left(\mathbf{P}_{\text{fg}}^\ell, \mathbf{Q}_{\text{fg}}^\ell\right) + (1 - \alpha) \cos\left(\mathbf{P}_{\text{bg}}^\ell, \mathbf{Q}_{\text{bg}}^\ell\right), \quad \alpha \in [0, 1]. \quad \text{(E.4)}$$

  *Intuition*: encourages higher agreement between support and query prototypes. *Caveat*: depends on the baseline mask $p_0$, which can be biased under shift.

- **Structure separation**

$$S_{\text{str}}(\ell) = 1 - \tfrac{1}{2} \left[ \cos\left(\mathbf{Q}_{\text{fg}}^\ell, \mathbf{Q}_{\text{bg}}^\ell\right) + \cos\left(\mathbf{P}_{\text{fg}}^\ell, \mathbf{P}_{\text{bg}}^\ell\right) \right]. \quad \text{(E.5)}$$

  *Intuition*: encourages foreground and background orthogonality in the query and support spaces. *Caveat*: measures feature geometry rather than final mask quality.

- **Texture and uncertainty complexity**

$$C(\ell) = \text{Var}\left(\mathbf{Q}_{\text{fg}}^\ell\right) + \text{Ent}(p_0), \quad \text{Ent}(p_0) = -\tfrac{1}{|\Omega|} \sum_x \left[ p_0 \log p_0 + (1 - p_0) \log(1 - p_0) \right]. \quad \text{(E.6)}$$

  Here $\text{Var}(\cdot)$ denotes the per dimension variance of query features relative to the corresponding prototype, weighted by $p_0$. *Caveat*: an indirect proxy that may penalize layers that are confident and correct.

**Gradient based proxies.** These rules favor layers with large loss sensitivity or sharp changes across adjacent layers.

**Gradient magnitude (Grad-Max).**

$$\ell_{\text{grad}}^\star = \arg\max_{\ell \in \mathcal{C}} \left\| \nabla_{\mathbf{F}_q^\ell} \mathcal{L}_{\text{base}} \right\|_2. \quad \text{(E.7)}$$

*Intuition*: select the layer to which the base loss is most sensitive. *Caveat*: residual paths and normalization in ViTs can amplify gradients in later layers, biasing the choice.

**Interlayer gradient change (Grad$\Delta$-Max).**

$$\ell_{\Delta\text{grad}}^\star = \arg\max_{\ell \in \mathcal{C}} \left\| \left\| \nabla_{\mathbf{F}_q^\ell} \mathcal{L}_{\text{base}} \right\|_2 - \left\| \nabla_{\mathbf{F}_q^{\ell-1}} \mathcal{L}_{\text{base}} \right\|_2 \right\|_2. \quad \text{(E.8)}$$

*Intuition*: detect transition points across adjacent layers. *Caveat*: still a gradient scale proxy, only weakly coupled to episode level decisions.

**Implementation notes.** All rules reuse a single forward pass of backbone activations. Gradient based proxies require one backward pass *without* parameter updates. The per episode cost is dominated by a single backpropagation through the frozen backbone.

### E.3  Task aligned HLS (ETR)

We select the routed layer by minimizing an episode level selection risk:

$$R_{\text{layer}}(\ell) = 1 - miou_{\text{sup}}(\ell), \qquad \ell^{\star}_{\text{HLS}} = \underset{\ell \in \mathcal{C}}{\arg\min}\, R_{\text{layer}}(\ell) = \underset{\ell \in \mathcal{C}}{\arg\max}\, miou_{\text{sup}}(\ell). \qquad \text{(E.9)}$$

Here $miou_{\text{sup}}(\ell)$ is computed within the episode by a leave one out procedure at layer $\ell$. Each support image is treated as a pseudo query and segmented using prototypes formed from the remaining supports, and the result is averaged over the $K$ supports.

*Rationale.* The criterion in equation E.9 directly measures episode level matching risk at the representation to be adapted, rather than optimizing a handcrafted surrogate. This makes it robust to layer level transfer variability and domain shift. In practice, HLS is parameter free, reuses the same forward features, and adds negligible overhead.

### E.4  Selector analysis and takeaway

**Why the three non episodic selectors underperform.** Table 4 compares per episode layer selectors with a frozen backbone. The *Static Max* rule blends three normalized cues and selects the layer with the largest $\alpha' S_{\text{sem}}(\ell) + \beta' S_{\text{str}}(\ell) + \gamma' C(\ell)$ (see Eqs. equation E.4 to equation E.6). These scores measure representation quality in feature space, including semantic agreement, structure separation, and texture or uncertainty, but they do not measure *task fit* for the episode. They lack episode level feedback and are therefore unstable across domains. Specifically, $S_{\text{sem}}$ inherits bias from the baseline mask $p_0$, $S_{\text{str}}$ rewards orthogonality that does not guarantee correct masks, and $C(\ell)$ can penalize layers that are confident and correct. The mixture weights $\alpha', \beta', \gamma'$ are domain specific. Consequently, Static Max averages 71.9 mIoU.

Gradient based proxies capture loss sensitivity rather than alignment. *Grad Max* selects the layer with the largest gradient norm (see Eq. (E.7)), and *Grad$\Delta$ Max* looks for sharp inter layer gradient changes (see Eq. (E.8)). In ViT backbones such as DINOv2 and DINOv3, blocks are architecturally homogeneous and connected by residual paths and layer normalization. This can cause gradients to grow toward the last blocks, so both rules tend to collapse to deep layers irrespective of the episode semantics. This Grad CAM style assumption therefore fails, and the selected layer often has the largest perturbation rather than being the most suitable for segmentation. These proxies correlate weakly with support and query matching quality and yield 73.1 and 73.2 mIoU on average.

**Why HLS (ETR) is better.** Our *HLS* uses a task aligned criterion that directly minimizes the episode level selection risk $\ell^{\star}_{\text{HLS}} = \arg\min_{\ell \in \mathcal{C}} \left(1 - miou_{\text{sup}}(\ell)\right)$ (see equation E.9). It performs a self prediction evaluation within the episode. Each support is treated as a pseudo query and is segmented using prototypes from the remaining supports, and the score is the support only mIoU at layer $\ell$. This provides dynamic, episode aware feedback aligned with the target objective, with low variance, no extra parameters, and negligible overhead. HLS reaches 76.7 mIoU, which is +4.8 over Static Max and +3.5 over the best gradient proxy. The gain is especially large on ISIC (from 48.2 to 73.6 mIoU, +25.4), and the gap widens on other VFM backbones such as DINOv2.

### E.5  DINOv2: component ablation (1-/5-shot) and takeaways

Table E.2. Component ablation on DINOv2 (average mIoU). $\Delta_{\text{V0}}$ denotes the improvement over the V0 baseline, and $\Delta_{\text{prev}}$ denotes the improvement relative to the row above. Best scores in bold.

| Setting | Avg. 1-shot | Avg. 5-shot | $\Delta_{\text{V0}}$ (1s / 5s) | $\Delta_{\text{prev}}$ (1s / 5s) |
| --- | --- | --- | --- | --- |
| V0 baseline (fusion=off, refine=off) | 57.03 | 68.49 | 0.00 / 0.00 | 0.00 / 0.00 |
| + HLS (enable fusion and routing) | 60.34 | 72.64 | +3.31 / +4.15 | **+3.31 / +4.15** |
| + PGR (Gaussian prior for attention) | 61.10 | 73.28 | +4.07 / +4.79 | +0.76 / +0.64 |
| + PAC (auto refine) | **62.58** | **73.42** | **+5.55 / +4.93** | +1.48 / +0.14 |

**Analysis.** The sequence *Select → Regularize → Calibrate* yields monotonic improvements. HLS provides the dominant gain by stabilizing the chosen adaptation layer for each episode. PGR reduces attention noise, such as spurious far field peaks, while preserving global coverage. PAC then corrects residual artifacts along thin boundaries and in low contrast regions. Gains are larger in the one shot

regime, where supervision is scarcer, which is consistent with the design intent. These results show that the hierarchical refinements generalize from DINOv3 to DINOv2 and to other VFMs, indicating effectiveness that is agnostic to the backbone.

**Practical remarks.** All selectors reuse cached features. HLS uses pseudo query scoring on the support only and therefore adds negligible overhead. PGR has no trainable parameters. PAC operates as a lightweight residual fusion and is gated automatically in five shot episodes. Consequently, the overall parameter and runtime budgets remain low while providing improvements that are aligned with the task.

## F    DISCLOSURE OF LARGE LANGUAGE MODEL (LLM) USAGE

We used large language models (LLMs) only to assist with writing. Specifically, LLMs were employed to polish wording, improve clarity, and refine the presentation (grammar, coherence, and flow) of certain sections. All scientific ideas, methodology, experiments, analyses, and conclusions were conceived and executed exclusively by the authors. LLM assistance was limited to language-related edits and suggestions. All outputs were reviewed and revised by the authors. The use of LLMs did not contribute to the research design, data collection, data analysis, or the intellectual content of the findings.

