# OpenReview forum: "HERA: Efficient Test-Time Adaptation for Cross-Domain Few-Shot Segmentation with Vision Foundation Models"
_ICLR.cc/2026/Conference — ICLR 2026 Conference Withdrawn Submission_

### Official Review · Reviewer_yZn5 · 2025-10-26

**Soundness:** 3
**Presentation:** 4
**Contribution:** 3
**Rating:** 4
**Confidence:** 4

**Summary:**

This paper aims to address the "fragility" of Vision Foundation Models (VFMs) when applied to Cross-Domain Few-Shot Segmentation (CD-FSS). The authors posit that this fragility stems from two primary challenges: 1) the extremely limited number of labeled exemplars leads to a high risk of overfitting during retraining, and 2) distribution shifts in the target domain cause internal representation mismatch and pronounced layer-wise sensitivity, which is episode-dependent.

To tackle this, the paper proposes HERA (Hierarchical Exemplar Representation Adaptation), a source-free, test-time adaptation (TTA) framework.

**Strengths:**

1. Precise Problem Definition: The paper astutely identifies "episode-dependent layer instability" as a primary failure mode for VFMs in CD-FSS, moving beyond typical explanations of overfitting or capacity.

2. Novel HLS Mechanism: HLS, guided by the ETR metric, is an elegant solution. It uses the task's own support data to guide model adaptation and is empirically proven (Table 4) to be far more effective than static heuristics or gradient-based proxies.

3.Good 5-shot Performance: The gain over SOTA in the 5-shot setting is substantial and demonstrates the method's effectiveness at leveraging VFM knowledge when $K>1$ samples are available.

4.Data and Parameter Efficiency: The method is source-free, which is a significant practical advantage.

**Weaknesses:**

1. The method's most significant drawback is its minimal improvement in the 1-shot setting. This severely undermines its claim as a general CD-FSS solution and implies the HLS/ETR mechanism is not robust for K=1.

2.As shown in Table 2, PGR and PAC contribute only ~1.2 mIoU combined, compared to +13.6 mIoU from HLS. The "three-stage" framework narrative feels oversold.

3. Using Vision Foundation Models for FSS is not a novel idea; several prior works have explored similar approaches. A comparison with more recent representative works, such as Matcher[1], is necessary. How does this training-free method perform on cross-domain few-shot segmentation tasks?

[1] Matcher: Segment Anything with One Shot Using All-Purpose Feature Matching. ICLR2024

**Questions:**

1.Regarding TTA latency: The adaptation is described as "per-episode." Does this mean that for two different query images (with their respective support sets) from the same target domain, HERA must run the full TTA process twice? Have you considered a "per-domain" adaptation strategy to amortize this cost?

2. How does it perform in industrial anomaly detection scenarios? Industrial anomaly detection often faces cross-domain and few-shot challenges; does the HERA method have advantages in such tasks?

---

### Official Review · Reviewer_nLCD · 2025-10-28

**Soundness:** 2
**Presentation:** 3
**Contribution:** 3
**Rating:** 4
**Confidence:** 4

**Summary:**

This paper introduces HERA (Hierarchical Exemplar Representation Adaptation), a source-free test-time adaptation framework for cross-domain few-shot segmentation (CD-FSS) with vision foundation models (VFMs). HERA follows a three-stage Select–Regularize–Calibrate process: it first selects the most transferable layer via an Exemplar Transfer Risk (ETR), then regularizes attention with spatial priors, and finally calibrates pixel-level predictions. By updating less than 2.7% of parameters, HERA achieves significant performance gains across multiple CD-FSS benchmarks.

**Strengths:**

1) The paper proposes a novel source-free test-time adaptation framework for cross-domain few-shot segmentation (CD-FSS).
2) HLS adaptively identifies the most informative (lowest-risk) VFM layer and restricts test-time updates to a small subset of its parameters, which is an insightful perspective.
3) Extensive experiments on four CD-FSS datasets demonstrate the effectiveness of the proposed method.

**Weaknesses:**

1) The experiments are conducted only on DINO, so it is unclear whether the conclusions hold or the method is effective for other VFMs.
2) The test-time adaptation (TTA) and selecting the lowest-risk layer may affect inference efficiency, which is not analyzed.
3) The ablation and comparative experiments are insufficient; the paper does not compare against direct VFM fine-tuning or baseline + TTA approaches.
4) Some relevant prior works are missing from both discussion and experimental comparison:\
[1] Adapter Naturally Serves as Decoupler for Cross-Domain Few-Shot Semantic Segmentation, ICML 2025\
[2] Self-Disentanglement and Re-Composition for Cross-Domain Few-Shot Segmentation, ICML 2025\
[3] Dual-Agent Optimization Framework for Cross-Domain Few-Shot Segmentation, CVPR 2025

5) The experimental results are not fully convincing: under the 1-shot setting, the method achieves the best performance on only one dataset, while it is suboptimal on the other three datasets.

**Questions:**

1) Could the authors clarify the performance of the VFM when fine-tuned on the target domain, including both full fine-tuning and LoRA-based fine-tuning?
2) What is the performance of performing test-time adaptation directly on the baseline (i.e., baseline + TTA)?
3) Does the inference process require two forward passes, one to select the lowest-risk layer and another after updating it? Given that HLS must be computed for each candidate layer and TTA involves backpropagation, does this substantially increase inference time, and how does the inference speed compare with other methods?
4) Are the proposed method and conclusions valid for VFMs other than DINO?
5) Could the authors provide the results of the 1-shot ablation experiments reported in Table 2?

---

### Official Review · Reviewer_k5Mi · 2025-10-30

**Soundness:** 3
**Presentation:** 2
**Contribution:** 3
**Rating:** 4
**Confidence:** 4

**Summary:**

The paper propose HERA, a lightweight framework to efficiently adapt frozen VFM features for cross-domain few-shot segmentation through a three-step "Select-Regularize-Calibrate" process. First, Hierarchical Layer Selection identifies the most informative intermediate VFM layer based on exemplar transfer risk. Second, Prior-Guided Regularization regularizes the per-head attention on the selected representation using an entropy-gated Gaussian prior. Third, Pixelwise Adaptive Calibration accurately corrects pixel-level prediction logits by integrating prototype similarity, attention propagation, and image appearance. Experiments show that HERA achieves gains in CD-FSS benchmarks with minimal computational cost.

**Strengths:**

1. The proposed method shows a reasonable degree of novelty. Cross-domain few-shot segmentation (CDFSS) is a meaningful and important area of study, and incorporating vision foundation models (VFMs) into this task is both attractive and aligned with the current research trend. The paper’s test-time adaptation method and the three-stage Select–Regularize–Calibrate framework are relatively novel within this field, representing a thoughtful methodological design rather than a simple engineering combination of existing modules.
2. The ablation studies in both the main text and the supplementary material are relatively complete and basically validate the method’s rationality.

**Weaknesses:**

1. The writing is poor, with some expressions that are not natural. While the paper sequentially describes the proposed framework, it lacks sufficient analysis of why each component is suitable for the CD-FSS task, which affects coherence and readability. Moreover, some parts—such as the definitions of ℓ_sim(x), ℓ_attn(x) and ℓ_img(x) (L337-L338)—are missing in the main text and only appear in the supplementary material, making the paper harder to follow. Some implementation details, like the masked average pooling used for prototype computation (L210–211) and how the few-shot head SSP integrates with HERA (L354–355) should also be included to make the paper more accessible, especially for readers unfamiliar with the CD-FSS. Some necessary details need to be added; see Questions* for specifics.
2. Some recent methods are missing in Table 1, such as GPRN and DATO:
Shi-Feng Peng, Guolei Sun, Yong Li, Hongsong Wang, and Guo-Sen Xie. Sam-aware graph prompt reasoning network for cross-domain few-shot segmentation. In Proceedings of the AAAI Conference on Artificial Intelligence, volume 39, pp. 6488–6496, 2025.
Zhaoyang Li, Yuan Wang, Wangkai Li, Tianzhu Zhang, and Xiang Liu. Dual-agent optimization framework for cross-domain few-shot segmentation. In Proceedings of the IEEE/CVF conference on computer vision and pattern recognition, pp. 9849–9859, 2025.
The results of these methods are better than the proposed approach.

**Questions:**

1. The leave-one-out strategy is clear for the 5-shot setting, but it seems inapplicable to the 1-shot case with only one sample. Could the authors clarify what is S^(-i) and how is this handled in the 1-shot experiments? (L207-L210)
2. The definition of 𝑈={12,…,23} is not very clear(L265). Does “in its neighborhood” refer to the previous and next layers of the best single layer ℓsingle?
3. In Figure 3, the quality of the heatmaps does not appear to consistently improve with deeper layers. For instance, layers 19–21 seem less informative than layers 14–15. Why Equation (3) introduces |ℓ - ℓ23| as a constraint?
4. Did the authors experiment with other vision foundation models beyond the DINO series, such as CLIP or SAM mentioned in the RELATED WORK section, to further verify the backbone-agnostic effectiveness? (L985–L986)
5. Table 1 does not include comparisons with several recent methods, which achieve better results than the approach presented in this paper; see Weaknesses* for specifics.

---

### Official Review · Reviewer_SEHw · 2025-10-31

**Soundness:** 4
**Presentation:** 3
**Contribution:** 4
**Rating:** 6
**Confidence:** 4

**Summary:**

The paper proposes the three-stage HERA framework to address VFM challenges in CD-FSS, and validates it on four datasets, with HERA (DINOv3) achieving 68.3/77.9 mIoU in 1/5-shot scenarios (surpassing SOTA) while being source-free and parameter-efficient.

**Strengths:**

1. Outstanding Innovation: Breaks source-data reliance and fixed-layer limits of existing CD-FSS methods, proposes a source-free, parameter-efficient VFM test-time adaptation paradigm, and pioneers the "layer selection–attention regularization–pixel calibration" chain; the ETR metric links task performance, outperforming Static-Max by 4.8 mIoU to solve VFM layer-wise transferability issues.
2. Rigorous Research Quality: Covers multi-domain datasets and multi-route methods in experiments, verifies module value via layered ablation; fine-tunes only 2.7% of parameters, with 0.735s 1-shot adaptation, balancing performance and deployment efficiency.
3. Clear Presentation: Follows "Problem–Method–Experiment–Conclusion" logic, standardizes core concepts and formulas; figures and text complement to intuitively convey framework flow and module necessity.
4. Significant Domain Impact: Theoretically expands test-time adaptation boundaries, proving lightweight optimization replaces full-layer fine-tuning; practically performs well in medical/remote sensing fields, aiding low-annotation multi-scenario segmentation.

**Weaknesses:**

1. Weak Theoretical & Generalization Analysis: Poorly explains the theoretical origin of VFM layer-wise transferability fluctuations and only validates HERA on DINOv2/DINOv3; suggest adding layer feature t-SNE analysis and tests on CLIP-ViT/SAM.
2. Suboptimal Experimental Details: Lacks hyperparameter transparency for reproduced methods (e.g., SSP) and domain shift quantification (e.g., MMD); suggest supplementing a hyperparameter table, code link, and MMD-based shift analysis.
3. Incomplete Module Design Details: Fails to clarify L23’s role in HLS fusion layers and uses fixed PAC residual weights; suggest adding L23 ablation experiments and testing dynamic PAC weight strategies.

**Questions:**

1. For HLS layer selection: Why is the candidate layer range 12–23, and does it need adjustment for other VFMs (e.g., CLIP-ViT)? Does ETR’s logic adapt to different VFM feature dimensions?
2. For PGR’s $\sigma_{loc}$/$\sigma_{glo}$: What’s the basis for their values? Is there dataset-specific sensitivity analysis (e.g., ISIC’s $\sigma_{loc}$)?
3. For PAC’s "auto 2/5" gating: Why is 2/5 optimal, and how to calculate gating’s "gain estimation"? Can threshold curves and gain formulas be added? 4. For extreme scenarios: How does HERA perform in extreme cross-domain/annotation noise? Are ETR and PGR robust here? Can verification experiments be supplemented?

---

### Note · Authors · 2025-11-14

I have read and agree with the venue's withdrawal policy on behalf of myself and my co-authors.